# Development of Highly Efficient Universal *Pneumocystis* Primers and Their Application in Investigating the Prevalence and Genetic Diversity of *Pneumocystis* in Wild Hares and Rabbits

**DOI:** 10.3390/jof10050355

**Published:** 2024-05-15

**Authors:** Liang Ma, Isabella Lin, Summer T. Hunter, Barbara Blasi, Patrizia Danesi, Christiane Weissenbacher-Lang, Ousmane H. Cisse, Jamie L. Rothenburger, Joseph A. Kovacs

**Affiliations:** 1Critical Care Medicine Department, National Institutes of Health Clinical Center, Bethesda, MD 20892, USA; isabellablin@gmail.com (I.L.); ousmane.cisse@nih.gov (O.H.C.); jkovacs@nih.gov (J.A.K.); 2Faculty of Veterinary Medicine, University of Calgary, Canadian Wildlife Health Cooperative (Alberta Region), Calgary, AB T2N 1N4, Canada; summer.hunter@ucalgary.ca (S.T.H.); jamie.rothenburger@ucalgary.ca (J.L.R.); 3Department of Biological Sciences and Pathobiology, Institute of Pathology, University of Veterinary Medicine Vienna, 1210 Wien, Austria; barbara.blasi@vetmeduni.ac.at (B.B.); christiane.weissenbacher-lang@vetmeduni.ac.at (C.W.-L.); 4Laboratory of Parasitology, Mycology and Medical Enthomology, Istituto Zooprofilattico delle Venezie, 35020 Legnaro, Italy; pdanesi@izsvenezie.it

**Keywords:** *Pneumocystis*, epidemiology, genetic diversity, phylogeny, hares, rabbits

## Abstract

Despite its ubiquitous infectivity to mammals with strong host specificity, our current knowledge about *Pneumocystis* has originated from studies of merely 4% of extant mammalian species. Further studies of *Pneumocystis* epidemiology across a broader range of animal species require the use of assays with high sensitivity and specificity. To this end, we have developed multiple universal *Pneumocystis* primers targeting different genetic loci with high amplification efficiency. Application of these primers to PCR investigation of *Pneumocystis* in free-living hares (*Lepus townsendii*, *n* = 130) and rabbits (*Oryctolagus cuniculus*, *n* = 8) in Canada revealed a prevalence of 81% (105/130) and 25% (2/8), respectively. Genotyping analysis identified five and two variants of *Pneumocystis* from hares and rabbits, respectively, with significant sequence divergence between the variants from hares. Based on phylogenetic analysis using nearly full-length sequences of the mitochondrial genome, nuclear rRNA operon and dihydropteroate synthase gene for the two most common variants, *Pneumocystis* in hares and rabbits are more closely related to each other than either are to *Pneumocystis* in other mammals. Furthermore, *Pneumocystis* in both hares and rabbits are more closely related to *Pneumocystis* in primates and dogs than to *Pneumocystis* in rodents. The high prevalence of *Pneumocystis* in hares (*P.* sp. ‘*townsendii*’) suggests its widespread transmissibility in the natural environment, similar to *P. oryctolagi* in rabbits. The presence of multiple distinct *Pneumocystis* populations in hares contrasts with the lack of apparent intra-species heterogeneity in *P. oryctolagi*, implying a unique evolution history of *P.* sp. ‘*townsendii*’ in hares.

## 1. Introduction

*Pneumocystis* is an opportunistic fungal pathogen capable of causing asymptomatic infection in healthy people but severe, even fatal, *Pneumocystis* pneumonia (PCP) if left untreated, in immunocompromised patients. In addition to humans, *Pneumocystis* is believed to infect most, and potentially all other mammals, including both wild and domesticated species [1], although *Pneumocystis* organisms infecting different mammal species are generally genetically distinct [2]. It is postulated that *Pneumocystis* species engage in a process of co-evolution with their respective hosts, which leads to a fairly strong host specificity [3,4]. Each *Pneumocystis* species typically infects only a single host species; multiple attempts to induce experimental cross-host species infection have consistently failed (reviewed by Durand-Joly et al. [5]).

*Pneumocystis* in rabbits (*Oryctolagus cuniculus*) was first described in 1916 [6] and has been formally named as *P. oryctolagi* in 2006 [7]. Rabbits have served as an important animal model for studying PCP, alongside mice (*Mus musculus*) and rats (*Rattus norvegicus*). They have contributed to the understanding of various aspects of *Pneumocystis* infection, including morphology [8,9], life cycle [10,11], transmission [12,13], genetic diversity [14,15,16], host immune response [17,18,19], pathogen–host interactions [18], and strategies for prevention and treatment [20,21].

Compared to other animals, rabbits exhibit distinct characteristics in their susceptibility to infection with *Pneumocystis*. Rabbits are the only animals studied to date that spontaneously develop natural overt *Pneumocystis* infection in the lungs with significant organism loads and histopathologic changes [7,22]. Studies of nonimmunosuppressed rabbits from commercial suppliers have shown that *Pneumocystis* infection commonly occurs around weaning (~1 month after birth), with the number of cyst forms comparable with that of corticosteroid-treated rabbits after weaning [22,23]. Concurrently, inflammatory histopathologic alterations develop in lung tissues, including vascular congestion, edema, infiltration of macrophages and eosinophils, and thickening of alveolar walls [22]. The infection usually resolves within 3–4 weeks. 

Another feature of *P. oryctolagi* noted by several studies is its closer ultrastructural similarity to primate *Pneumocystis* than to rodent *Pneumocystis*, including markedly fewer surface membrane protrusions in trophic forms (known as filopodia), and fewer and smaller membrane-associated electron-dense cytoplasm granules in the former than the latter (reviewed by Dei-Cas et al. [7]). Furthermore, recent genome analysis has demonstrated that *P. oryctolagi* is phylogenetically more closely related to primate *Pneumocystis* than rodent *Pneumocystis* despite its host species, rabbits, being more closely related to rodents than primates [24].

Hares, belonging to the same family as rabbits (Leporidae), are widely distributed worldwide, though much less studied than rabbits for infection with *Pneumocystis.* There are only four reports describing morphological detection of *Pneumocystis* in wild hares, including brown hares (*Lepus europaeus*) and mountain hares (*Lepus timidus*) in Europe [25,26,27,28]. In these studies, there were no microscopic changes in the lungs in most of infected hares, including one with a high load of *Pneumocystis* organisms [25].

*Pneumocystis* still cannot be cultured reliably in vitro, which has greatly hampered the research on its basic biology and disease process [3]. While laboratory animals, including rabbits, mice, and rats, have been widely used as models to understand human disease, the results do not always reflect natural infection. Thus, it is important to investigate *Pneumocystis* in wildlife to better understand the disease in humans. 

In this study, we capitalized on the availability of lung specimens from deceased wild hares (white-tailed jackrabbits [*Lepus townsendii*]) and feral European rabbits (*Oryctolagus cuniculus*) presumed to have succumbed to road incidents in Calgary, Alberta, Canada. The objectives of this study were to develop universal *Pneumocystis* primers with improved species coverage and PCR amplification efficiency, and subsequently to apply these primers to investigate the prevalence and genetic diversity of *Pneumocystis* organisms in these free-living animals. To simplify description and facilitate future referencing, *Pneumocystis* identified from the hares (*L. townsendii*) in this study is referred to as *P.* sp. ‘*townsendii*’ following the reference of Schoch et al. [29]. 

## 2. Materials and Methods

### 2.1. Sample Source and DNA Extraction

This study involved 135 hares and 8 rabbits from Calgary, Alberta, Canada, collected between October 2019 and July 2020 (Appendix A). Animal carcasses were initially received by either the City of Calgary’s Road Services or the Calgary’s Wildlife Rehabilitation Society. Animals from the rehabilitation facility were held in captivity for more than 72 h prior to death or euthanasia. Carcasses were also excluded if they had severe decomposition or were missing a significant portion of the organs (particularly lungs). For each carcass, the submitter provided data on geographic location and date of retrieval. All carcasses were stored in a −20 °C freezer before being transported to the laboratory at the University of Calgary for detailed autopsy examinations, which included the assessment of sex, maturity status, body mass, and lesions. Two small pieces of lung tissues were collected from each animal and kept at −80 °C until they were shipped on dry ice to the laboratory at NIH, Bethesda, MD, USA. 

Host species were identified morphologically by experienced veterinary professionals. Additionally, selected animals (positive in *Pneumocystis* PCR tests) were further confirmed by PCR and sequencing of the host mitochondrial cytochrome B (*cytb*) gene using primers CTB.f3 and CTB.r3 (Appendix A). The host *cytb* sequences from all hares and rabbits sequenced were identical to reference sequences for hares (*L. townsendii*, GenBank accession nos. NC_024041.1 and HQ596485.1) and rabbits (*O. cuniculus*, NC_001913.1), respectively. 

Genomic DNA was extracted from lung tissues (~100 mg each animal) using the QIAamp Fast DNA Tissue Kit (Qiagen) and eluted in ~50 μL. DNA extracts were quantitated by NanoDrop spectrophotometer (Thermo Fisher Scientific, Waltham, MA, USA). 

### 2.2. Design and Testing of Universal Primers for the Pneumocystis Genus 

Multiple pairs of primers were selected from highly conserved regions of the full-length dihydropteroate synthase (*dhps*), nuclear rRNA operon (rDNA), mitochondrial large-subunit rRNA gene (mtLSU), and mitochondrial small-subunit rRNA gene (mtSSU) in *Pneumocystis* from 7 animal species reported previously [24,30,31]. We chose these targets because they have been widely used for the detection of *Pneumocystis* and/or for the investigation of its genetic diversity and phylogeny across various mammal species. 

The sequences of these new primers, along with previously reported primers [32,33,34,35,36], are listed in Appendix A. Alignment of selected mitochondrial and rDNA primers with targeted sequences from different *Pneumocystis* species (including references [33,34,37,38,39]) are shown in Appendix A.

The amplification efficiencies of these primers, along with previously reported primers, were determined using a pool of DNA extracts from 20 hares positive for *Pneumocystis* in a preliminary screening by PCR targeting mtSSU with primers SSU.f1 and SSU.r1 [35]. 

PCR was performed in a 50 µL volume containing ~500 ng genomic DNA, 0.25 M of each primer, and 25 µL of LiTaq Plus Mix (LifeSct LLC, Rockville, MD, USA) using a touch-down thermocycling program as follows: 94 °C for 2 min; 10 cycles of 94 °C for 30 s, 65 °C for 60 s with 1.5 °C decrease per cycle, and 72 °C for 2 min; 30 cycles of 94 °C for 15 s, 50 °C for 30 s, and 72 °C for 1.5 min; and a final extension at 72 °C for 5 min. PCR products were examined in 1% or 1.2% E-gel (Thermo Fisher Scientific). The PCR products with a single, strong band with the expected size were purified and sequenced directly by Sanger sequencing. 

The amplification efficiency of primer pairs was compared based on the number, size and density of the DNA bands in the PCR products visualized on agarose gels. The presence of a single, strong band of the expected size indicates an efficient amplification. The primer pairs showing a high efficiency, including rns.f8-rns.r13 for mtSSU, rnl.f10-Pu3Y.f for mtLSU, PK.f2-dhps.r6 for *dhps*, and ITS1.f-FUN-ITS4 for ITS1-5.8S-ITS2 (nuclear internal transcribed spacer 1-5.8S rRNA-internal transcribed spacer 2), were chosen for the prevalence or genotyping study as described below. In addition, the primer pairs rns.f8-rns.r13 and rnl.f10-Pu3Y.f were further tested for their efficiency in detecting *Pneumocystis* from an additional 14 mammal species, with details about the samples and references [24,30,32,35,37,40,41,42,43,44] listed in Appendix A. PCR was conducted using the same conditions described above. All PCR products were purified and sequenced directly by Sanger sequencing to confirm the sequence identity. 

### 2.3. Determination of the Prevalence of Pneumocystis Infection in Hares and Rabbits by PCR 

We chose to determine the prevalence of *Pneumocystis* by PCR targeting the mitochondrial genes due to their presence of multiple copies per organism, which is expected to be more sensitive than PCR targeting single-copy nuclear genes [45]. All DNA specimens were initially amplified in parallel for mtSSU and mtLSU using the primer pairs rns.f8-rns.r13 and rnl.f10-Pu3Y.f, respectively. PCR was conducted using the same conditions described above. For samples showing inconsistent results between the mtSSU and mtLSU PCR tests, nested PCR was performed using primer pairs rns.f6b-rns.r13 and rnl.f10-LSU.r1 (Appendix A), respectively. Universal PCR precautions were taken to avoid potential contamination, including the use of separate workplaces for pre-PCR and post-PCR steps, aliquoting of primers and other reagents for one-time use, setting-up of reaction mixtures in a dedicated hood, and inclusion of negative controls in each experiment. 

All positive PCR products were purified and sent to a commercial sequencing facility (Quintara Biosciences, Frederick, MD, USA) for Sanger sequencing. 

### 2.4. Genetic Divergence Analysis of Pneumocystis in Hares and Rabbits by Multi-Locus PCR and Sequencing 

To investigate the genetic divergence of *Pneumocystis*, we performed PCR and sequencing of the ITS1-5.8S-ITS2 and *dhps* genes in addition to the mtSSU described above. Only positive samples detected by mtSSU-PCR were used for this analysis since single-copy nuclear genes are unlikely to be amplified from samples negative by mtSSU-PCR. Primer pairs PC16S.f4-16SrRNA.r1 and PK.f2-dhps.r6 (Appendix A) were used to amplify the ITS1-5.8S-ITS2 and *dhps* genes, respectively, as conducted using the conditions described above. Samples showing negative results or weak products in PCR with primer pair PK.f2-dhps.r6 were subjected to nested PCR with primer pair PK160-dhps.r6 (~470 bp). All positive PCR products were purified and sequenced commercially by direct Sanger sequencing. 

### 2.5. Sequencing of the Mitochondrial Genome (mtDNA) and Nuclear rRNA Operon (rDNA) of P. sp. ‘Townsendii’

For two hares (HR20-009 and HR20-090), the nearly complete mtDNA of *Pneumocystis* (~21 kb without the non-coding regions in both ends) was amplified in 7 overlapping fragments using primer pairs rnl.r8-LSU.f1, rnl.r12-rnl.f1c, rnl.r4-nad5.r6, nad5.f6-cob.r8, cob.f8-rns.f8, rn3.r3-cox1.f5, and cox1.r5-mt.r103c (Appendix A). In addition, the nearly full-length rDNA (~5.5 kb) was amplified in two overlapping fragments using primer pairs PCP18S.f3-16SrRNA.r1 and Nu26S.f9-PCP28S.r1 (Appendix A). All these primers were selected from highly conserved regions of the full-length mtDNA and rDNA of *Pneumocystis* from 7 animal species reported previously [24,30,31].

PCR was performed using total genomic DNA and the LiTaq Plus Mix (LifeSct LLC, Rockville, MD, USA) with a touch-down thermocycling program as follows: 94 °C for 2 min; 10 cycles of 94 °C for 30 s, 65 °C for 60 s with 1.5 °C decease per cycle, and 72 °C for 7 min; 30 cycles of 94 °C for 15 s, 50 °C for 30 s, and 72 °C for 7 min; and a final extension at 72 °C for 10 min. All PCR products were purified. After partial Sanger sequencing to confirm the presence of unique *Pneumocystis* sequences, all PCR products from each animal were pooled together as one amplicon mixture for next-generation sequencing (NGS). NGS was performed commercially in an Illumina HiSeq platform using a 150-base paired-end library. 

Illumina data were assembled using SeqMan NGen (version 14.1.0.118, DNASTAR, Madison, WI, USA) under default conditions. The resulting contigs were oriented using Sequencher by aligning to the *P. oryctolagi* mtDNA and rDNA reference sequences (GenBank accession nos. MT726213.1 and MT780543.1). The final assemblies were re-aligned to Illumina raw reads using SeqMan NGen to check for any potential assembly errors as well as intra-species single nucleotide polymorphisms (SNPs). 

### 2.6. Gene Annotation, Sequence Similarity, and Evolutionary Distance Estimation 

The *Pneumocystis* mtDNA assembly was first annotated using the MFannot tool at http://megasun.bch.umontreal.ca/cgi-bin/mfannot/mfannotInterface.pl (accessed 1 October 2023) [46]. All annotated genes were reviewed and compared with the homologs in *P. murina* and *P. carinii* mtDNAs, and by blast against the NCBI database. tRNA genes were further evaluated using the tRNAscan-SE server at http://lowelab.ucsc.edu/tRNAscan-SE (accessed 21 September 2023) [47]. 

Sequence alignment was conducted using MacVector under default conditions of the MUSCLE program (version 18.6.1, MacVector, Inc., Apex, NC, USA). Sequence identity and similarity were determined using the Matrix mode of MacVector. Genetic distances were calculated using MEGA (version 11.0.13 [48]), with default setting and the Jukes–Cantor model of the pairwise distances algorithm. 

### 2.7. Phylogenetic Analysis 

DNA sequences were collected, checked for proper orientation using emboss revseq, and aligned using Clustal Omega [49]. All 18 mitochondrial genes were concatenated into one super alignment containing 19,052 nucleotide sites with 50.9% of constant sites and 5469 parsimony informative sites. The maximum likelihood phylogeny was inferred using IQ-TREE [50] with the GTR+F+I+G4 as the best fit model according to the Bayesian information criterion (BIC). The alignment of the *dhps* sequences contained 1326 nucleotide sites (62.36% constant sites; 278 distinct site patterns, best-fit model according to BIC: HKY+G4). The rDNA alignment contained 7062 nucleotide sites (74.90% constant sites; 990 parsimony informative sites, best-fit model according to BIC: GTR+F+I+G4). 

Branch tests were performed using 1000 replicates with the SH-like approximate likelihood ratio test (SH-aLRT) [51] and the ultrafast bootstrap (UFBoot) approximation [52]. 

### 2.8. Statistical Analysis

All statistical analyses were conducted in R Studio (version 2023.12.1+402). *Pneumocystis* positive rates determined by mtSSU-PCR and mtLSU-PCR were compared by a Chi square test. The association of *Pneumocystis* positive rates with the sample collection date and animal sex was assessed by the Pearson’s product–moment correlation test. A *p* value < 0.05 was considered statistically significant. 

## 3. Results

### 3.1. Universal Pneumocystis Primer Design

For mtSSU, out of a total of ten primer pairs tested, four showed the best performance with a single, strong band with the expected size (~600–670 bp. Figure 1A). All other primer pairs yielded multiple bands. The primer pair rns.f8-rns.r13 was chosen for the prevalence studies due to its shorter amplicon size (~600 bp), which presumably has a higher amplification efficiency than the other primer pair with a larger amplicon size (~670 bp). 

For mtLSU, almost all new primer pairs designed in this study showed a single, strong band with the expected size of ~430–750 bp (Figure 1B). The two previously reported primer pairs yielded multiple bands. The pair rnl.f10–Pu3Y.f was chosen for the prevalence studies due to its suitable amplicon size (~500 bp) and significant overlap with the amplicon sequences of the previously reported primer pairs.

Both primer pairs rns.f8-rns.r13 and rnl.f10–Pu3Y.f showed strong amplification when applied to samples from 14 other animal species, including some formalin-fixed, paraffin-embedded lung tissue samples with DNA highly degraded (Figure 2A,B). All PCR products were confirmed by direct Sanger sequencing (Appendix A).

For the ITS1-5.8S-ITS2 and *dhps* genes, out of 9 and 11 pairs of primers tested, pairs ITS1.F-Fun.ITS4 and PK.f2-dhps.r8 showed excellent amplification efficiency, respectively (Figure 1C,D), and were used to assess the genetic diversity at these two loci in hare and rabbit samples. In addition, they were also successfully used to amplify samples from all 14 other mammal species except for ferrets, pigs, goats, shrews, river rats and cats, which were not tested due to low DNA concentrations or quality (extensive degradation).

### 3.2. Prevalence of Pneumocystis in Hares and Rabbits

Out of the 130 hares, 99 (76%) and 95 (73%) samples were positive for the mtSSU and mtLSU genes, respectively, after a single round of PCR. Following semi-nested PCR on samples showing different results for these two genes, an additional 2 and 4 samples became positive for mtSSU and mtLSU, giving rise to a total of 101 (78%) and 99 (76%) positive samples, respectively. There was no significant difference between these positive rates (*p* = 0.883). There were a total of 105 (81%) samples positive for either genes and 25 (19%) samples negative for both genes. There were four samples positive for mtLSU but negative for mtSSU, while there were six samples positive for mtSSU but negative for mtLSU. In determining prevalence, any sample testing positive for either gene was considered as a positive case, i.e., 105 out of 130 animals (80.8%).

Based on direct Sanger sequencing, all PCR products contained sequences with the highest identities (88–99%) to *P. oryctolagi* sequences in the GenBank database according to nucleotide BLAST (https://blast.ncbi.nlm.nih.gov/Blast.cgi) (accessed 6 July 2023). Sequence analysis of mtSSU PCR products revealed five variants (v1 to v5) of *P.* sp. ‘*townsendii*’, with variant 1 being the most common (91%), followed by variant 2 (6%). All other variants were infrequent, each comprising 2% or less of the total (Figure 3). Only three of these variants (variants 1, 2 and 3) could be amplified by mtLSU PCR from 92 (70.8%), 5 (3.8%) and 2 (1.5%) hares, respectively (Appendix A). 

The prevalence of *Pneumocystis* in hares (regardless of variants) was not associated with the sample collection date (*p* = 0.2) or animal sex (*p* = 0.8). Potential associations with variants were not assessed due to the small number of samples for variants 2 to 5. 

Out of the eight rabbits, only two (25%) were positive for both mtSSU and mtLSU after a single round of PCR. All others were negative for either gene; none of them were positive in nested-PCR for either gene. 

### 3.3. Genetic Variation of Pneumocystis in Hares and Rabbits

Among the five variants of *P.* sp. ‘*townsendii*’ determined at the mtSSU gene (~600 bp), the nucleotide divergence varied from 0.2% to 9.4%. Among the three variants at mtLSU (~500 bp), the nucleotide divergence varied from 0.4% to 6.7% (Appendix A), corresponding to the three variants at mtSSU with a divergence of 0.2% to 2.3%. 

For the ITS1-5.8S-ITS2 locus (~610 bp), only variants 1 and 2 were amplified from fifty and two hares, respectively, with SNPs and indels presented in six positions (1.3% divergence) between these two variants (Appendix A). This locus could not be amplified from hares with variants 3 to 5. Similarly, for the *dhps* gene (~1.3 kb), only variants 1 and 2 were amplified from 32 and 3 hares, respectively, with 21 SNPs (1.6% divergence) noted between them (Appendix A). This locus could not be amplified from hares with variants 3 to 5. The failure in PCR amplification of the ITS1-5.8S-ITS2 and *dhps* loci in 52/94 (55%) mtSSU-postive samples can be explained by the lower copy number per genome of these loci and thus a lower sensitivity in PCR amplification compared to the mitochondrial gene discussed above. 

To provide a more accurate estimation of the genetic similarity of *Pneumocystis* from hares and other mammals, we obtained the nearly complete sequences of mtDNA, rDNA, and *dhps* of *P.* sp. ‘*townsendii*’ variants 1 and 2 (each variant from one hare). These two variants showed a similarity of 94.7%, 99.2%, and 98.4% at mtDNA, rDNA, and *dhps*, respectively. Compared to *Pneumocystis* species from other animal species, the two variants of *P.* sp. ‘*townsendii*’ showed the highest similarity to *P. oryctolagi* at all three loci, and the lowest similarity to *P. carinii* at both rDNA and mtDNA and *P. wakefieldiae* at *dhps* (Appendix A). These similarity levels are in good agreement with the genetic distances estimated from the same genetic regions (Appendix A). Of note, there were a total of 13 SNPs in the nearly full length *dhps* open reading frame (818 bp) between the two variants in hares, which resulted in amino acid changes in four codons (though not in the corresponding region in *P. jirovecii* containing missense mutations associated with sulfa exposure) (Appendix A). 

In the two infected rabbits, mtSSU-PCR yielded an identical sequence (605 bp), which harbored three SNPs compared to the *P. oryctolagi* reference sequence in GenBank (NC_060319.1). However, based on mtLSU-PCR, each rabbit harbored a unique *P. oryctolagi* sequence (494 bp), with one and three SNPs, respectively, compared to the reference sequence in GenBank (NC_060319.1). The ITS1-5.8S-ITS2 and *dhps* genes could not be amplified from either rabbit, presumably due to the low *P. oryctolagi* load in these samples. 

### 3.4. Phylogeny of Pneumocystis in Hares and Rabbits 

Based on phylogenetic analysis using concatenated sequences of the mtDNA core genes, *Pneumocystis* in both hares and rabbits were more closely related to each other than to *Pneumocystis* in other mammals. However, both were more closely related to primate and dog *Pneumocystis* than to rodent *Pneumocystis* (Figure 4). The tree topology inferred from mtDNA was congruent with that from *dhps* and rDNA except for the position of *P. wakefieldiae*, which exhibited an alternate placement with low bootstrap support in the rDNA-based tree, appearing more closely related to *P. carinii* than *P. murina* (Appendix A). This contradicts its placement in the mtDNA and dhps-based trees, where it was positioned closer to *P. murina* than *P. carinii*, albeit with strong bootstrap support. The tree topology from all three genomic loci was consistent with the genetic distances estimated from these loci (Appendix A). 

## 4. Discussion

Better understanding the epidemiology of *Pneumocystis* requires investigation of a broad range of animal species from diverse environments. Given the substantial genetic variation between *Pneumocystis* from different animal species, molecular identification of *Pneumocystis*, particularly in previously unexamined animal populations, necessitates the use of assays with high sensitivity and specificity. To this end, we have developed multiple universal *Pneumocystis* primers targeting different genetic loci with high amplification efficiency. With these primers we investigated the prevalence of *Pneumocystis* in free-living hares and rabbits in Canada and found a high prevalence of *Pneumocystis* infection, particularly in hares. Genotyping analysis identified five and two variants of *Pneumocystis* from hares and rabbits, respectively. Phylogenetic analysis demonstrated that *Pneumocystis* in hares and rabbits are more closely related to each other than either are to *Pneumocystis* in other mammals. Furthermore, *Pneumocystis* in both hares and rabbits are more closely related to *Pneumocystis* in primates and dogs than to *Pneumocystis* in rodents. This is the first report on the genetic sequences and molecular epidemiology of *Pneumocystis* in hares.

This study offers multiple new primer sets with broad coverage of diverse *Pneumocystis* species and high amplification efficiencies, which are expected to facilitate the investigation of *Pneumocystis* epidemiology and genetic diversity in various animal species. Currently, the majority of epidemiological studies of *Pneumocystis* infection in nonhuman animals have relied on the use of the universal mtLSU primers reported in the 1990s, which were originally derived from *P. jirovecii* and *P. carinii* [34,38,39,53]. These primers have made significant contributions to the identification, genotyping, and phylogeny analysis of *Pneumocystis* from almost all host species examined to date. Although there are reports of mismatches of these primers with the targets, even among different *P. jirovecii* isolates [30,33], there have been no studies directly evaluating the efficiency and sensitivity of these and other primers in detecting *Pneumocystis* in humans or nonhuman species. In this study, we aligned these primers with currently available mitogenomes from multiple *Pneumocystis* species and observed sequence mismatches in one or multiple positions in each of these primers (Appendix A). These mismatches, particularly those located near 3′ end of the primers, likely reduce the PCR amplification efficiency, leading to low or no product yield [54]. To circumvent this issue, we designed new universal primers with broad species coverage and high amplification efficiency not only for mtLSU but also for mtSSU and nuclear genes *dhps* and ITS1-5.8S-ITS2 (Figure 1 and Figure 2). 

Both mtLSU and mtSSU primers described in this study are suitable for prevalence studies, given their presence in multiple copies per cell, as well as genetic diversity and phylogeny studies, owing to substantial sequence diversity in the internal regions of these primers. Both *dhps* and ITS1-5.8S-ITS2 primers are suitable for genetic diversity and phylogeny studies, rather than prevalence studies, as their targets are present in a single copy per cell, resulting in lower sensitivity compared to mitochondrial primers. This is consistent with the PCR amplification failure for these primers observed in ~55% of mtSSU-positive samples in this study. The targeted regions by all new primers described in this study have significant overlaps with those by previously reported primers, thereby allowing comparison of the sequences amplified from new samples by these primers with existing sequences. 

Of note, the primer Pu3Yf for mtLSU was adopted from Pu3for [33] by incorporating a degenerate nucleotide Y at the 11st position (from the 3′ end) based on the sequence alignment shown in Appendix A. The C–T mismatch with *Pneumocystis* from squirrels was not noted until its mitogenome sequence became available recently (Appendix A). It is unknown how this mismatch will affect the amplification of squirrels or others with the same target sequence as squirrels in a large-scale study, while this primer pair showed strong amplification of *Pneumocystis*-infected squirrel samples in this study (Figure 2B), implying a minimal adverse effect of the C–T mismatch on the PCR product yield as previously documented [54]. If any adverse effect occurs due to this mismatch, alternative primer pairs can be chosen from our list, e.g., rnl.f10-LSU.r1 and LSU.f1-LSU.r1 (Figure 1 and Appendix A). Nevertheless, we recommend testing PCR amplification efficacy when using any primers in this study for studies of *Pneumocystis* in all other animals, especially those having not been previously tested for *Pneumocystis* infection. Given the very low ratio of *Pneumocystis* DNA in samples from immunocompetent host species, one concern arises that the primers may bind to host DNA and such binding is apt to be affected by the sequence variation between different host species. Hence, it is imperative to optimize PCR conditions when adapting primers reported by others.

With the new universal mitochondrial primers, we documented, for the first time, a high prevalence of *P.* sp. ‘*townsendii*’ in 81% of white-tailed jackrabbits (*L. townsendii*). This prevalence is substantially higher than that reported in mountain hares (*L. timidus*, 17%) [27] and brown hares (*L. europaeus*, 11–20%) [25,26,27]. The higher prevalence in our study can potentially be attributed to the use of highly efficient PCR assays, as opposed to the insensitive microscopic detection of stained lung tissues used in the previous studies [25,26,27]. The very small sample size for rabbits in this study precludes any meaningful conclusion regarding the prevalence. The prevalence of *P.* sp. ‘*townsendii*’ appears to be higher than that reported from wild rabbits (58%) [55] and similar to that reported from commercially bred domestic rabbits (80–100%) [22], both also determined by PCR. This prevalence is also similar to those reported from some other wild and domesticated animals, including raccoon dogs (78%, [33]), Valais shrews (80%, [37]), and various rodents (78–93%, reviewed by Weissenbacher-Lang et al. [1]). We did not assess the lungs for microscopic lesions associated with infection or the presence of *Pneumocystis* organisms due to freeze–thaw artifact and autolysis of the lung tissues.

Another interesting finding in this study is the identification of five variants of *P.* sp. ‘*townsendii*’ with considerable sequence divergences at mtSSU (0.2–9.4%. Figure 3). Particularly, variant 5 displayed a 7.7–9.4% divergence with the other four variants, a level seemingly reaching beyond the inter-strain divergence levels in known *Pneumocystis* species [24]. This raises the possibility of the presence of different *Pneumocystis* species in hares; further studies of additional genomic regions are needed to address this possibility. A similar finding has been observed in various other wild and domestic animals, particularly brown rats [56], Rhesus macaques [57], dogs [24], and pigs [58,59], but not in rabbits. Despite the presence of multiple *Pneumocystis* variants in this hare population, there was no evidence of mixed sequence populations representing co-infection with multiple variants within any hare samples based on the chromatographs of direct Sanger sequencing for all positive samples, as well as NGS reads for PCR products of the nearly full mtDNA and rDNA from the two positive samples (HR20-009 and HR20-090). While this observation requires validation through NGS or other high-throughput methodologies and using additional samples from broad geographical origins, it potentially indicates a reduced prevalence of *Pneumocystis* co-infection within the same hares compared to humans (reviewed by Ma et al. [2]) and many wild and captive animals, such as rats [56,60] and Rhesus macaques [57], which have demonstrated a high prevalence of co-infection based on NGS or even direct Sanger sequencing of PCR products. 

Our phylogenetic analysis shows that *Pneumocystis* in hares and rabbits are more closely related to each other than either are to *Pneumocystis* in other mammals (Figure 4), as expected based on the prevailing co-evolution belief within the *Pneumocystis* genus. However, *Pneumocystis* in both hares and rabbits are more closely related to *Pneumocystis* in primates and dogs than to *Pneumocystis* in rodents. This is consistent with the results of previous phylogenetic analyses of *P. oryctolagi* and other *Pneumocystis* species based on whole nuclear genome data, but contradicts the co-evolution hypothesis since it has long been believed that Lagomorpha (including hares and rabbits) are more closely related to rodents than to primates [61,62,63]. The discordant co-evolution of *P. oryctolagi* with its host species suggests the occurrence of host switching events in their evolutionary past. However, the evolutionary relationships among rodents, lagomorphs, and primates remain controversial, as there is also evidence suggesting a closer relationship between lagomorphs and primates than to rodents [64,65,66]. 

Despite its close phylogenetic relationship to and similarly high prevalence as *P. oryctolagi*, *P.* sp. ‘*townsendii*’ differs from *P. oryctolagi* in its presence of multiple distinct populations with significant sequence divergence as demonstrated in this study and the lack of apparent pathological changes in the lungs in most of the infected hares as previously reported. These findings may reflect a unique evolutionary trajectory of *P.* sp. ‘*townsendii*’ compared to *P. oryctolagi*.

## 5. Conclusions

We have developed highly efficient universal primers targeting multiple mitochondrial and nuclear genes of the *Pneumocystis* genus. With these primers we investigated the prevalence and genetic diversity of *Pneumocystis* in free-living hares and rabbits in a Canadian city and found a high prevalence of *Pneumocystis* infection with multiple genetically distinct variants, particularly in hares. Despite its close phylogenetic relationship to *P. oryctolagi* in rabbits, *P.* sp. ‘*townsendii*’ in hares may have experienced a unique evolutionary history. To address this possibility, further studies are warranted using a large-scale genomic investigation encompassing geographically and temporally diversified samples. Additionally, quantitative detection in conjunction with histopathological examination of fresh samples would help characterize the intensity of infection and distinguish *Pneumocystis* pneumonia from colonization or subclinical infection. 

## Figures and Tables

**Figure 1 jof-10-00355-f001:**
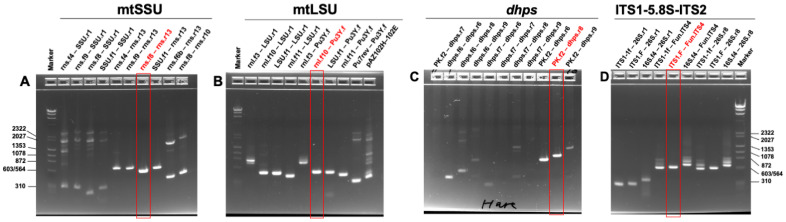
Efficiencies of various combinations of universal *Pneumocystis* primers in amplifying hare DNA samples for (**A**) the mitochondrial small-subunit rRNA gene (mtSSU), (**B**) mitochondrial large-subunit rRNA gene (mtLSU), (**C**) dihydropteroate synthase gene (*dhps*), and (**D**) nuclear internal transcribed spacer 1-5.8S rRNA-internal transcribed spacer 2 (ITS1-5.8S-ITS2). Indicated above each lane are primer pairs, with their sequences provided in Appendix A. PCR products were separated on 1.2% E-Gel with SYBR Safe DNA Gel Stain (ThermoFisher Scientific). Highlighted in red are primer pairs that showed strong amplification efficiencies and were used to investigate the prevalence and genetic divergence of *Pneumocystis* in hares and rabbits in this study. Marker is Lambda DNA Hind III mixed with phiX 174 DNA Hae III (ThermoFisher Scientific), with the sizes (bp) indicated for selected bands. The label ‘hare’ on the bottom of panel C was used to distinguish this gel from others photographed in the same batch.

**Figure 2 jof-10-00355-f002:**
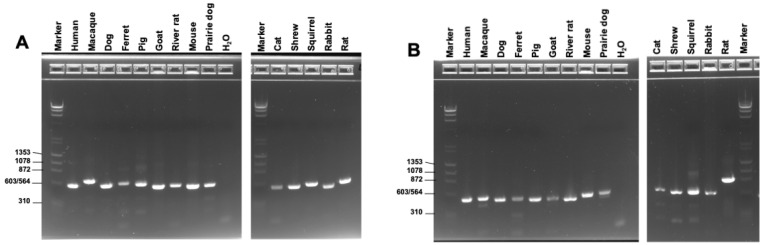
Application of new universal *Pneumocystis* primers to amplify *Pneumocystis* from 14 different animal species. (**A**) Amplification with the mtSSU primer pair rns.f8-rns.r13, selected from Figure 1A. (**B**) Amplification with the mtSSU primer pair rnl.f10-Pu3Y.f, selected from Figure 1B. Indicated above each lane are the animal species from which *Pneumocystis* samples were obtained. Details on the sample source are provided in Appendix A. The marker is the same as in Figure 1.

**Figure 3 jof-10-00355-f003:**
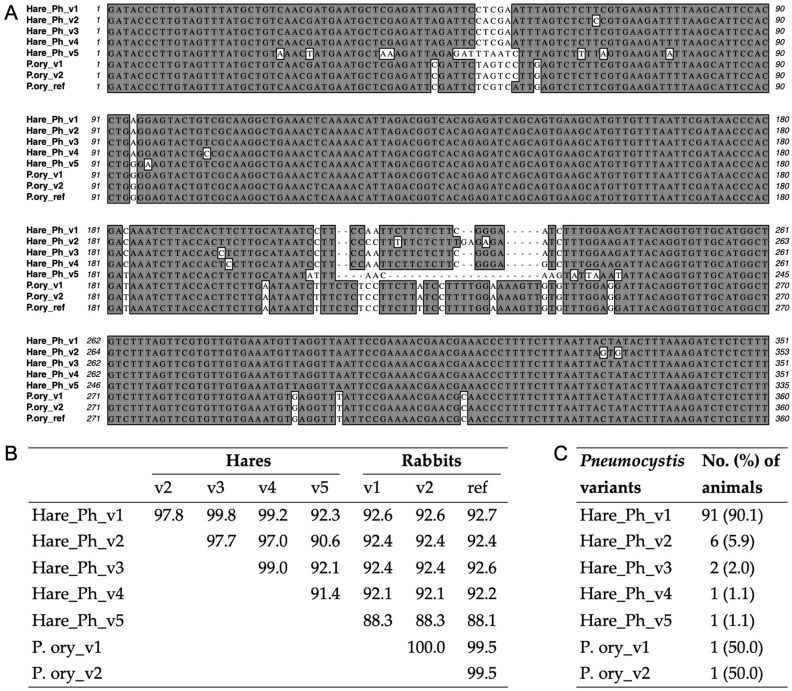
Identification of multiple *Pneumocystis* variants in hares and rabbits from Canada based on mtSSU. (**A**) Partial mtSSU sequences of *Pneumocystis* variants aligned with *P. oryctolagi* reference sequence (P.ory_ref, from MI, USA. GenBank accession no. MT726213). Identical nucleotides among multiple variants are highlighted in dark shadow. Of note, the two *P. oryctolagi* variants identified in this study are represented by P.ory_v1 and P.ory_v2, which have identical mtSSU sequences, as shown here, but have different mtLSU sequences, as shown in Appendix A. (**B**) mtSSU sequence identity (%) between *Pneumocystis* variants based on alignment of the full-length sequences of the PCR products (GenBank accession nos. PP477339-PP477344). (**C**) Frequencies of *Pneumocystis* variants in infected animals. The five variants of *P.* sp. ‘*townsendii*’ identified in this study are represented by Hare_Ph_v1 to Hare_Ph_v5.

**Figure 4 jof-10-00355-f004:**
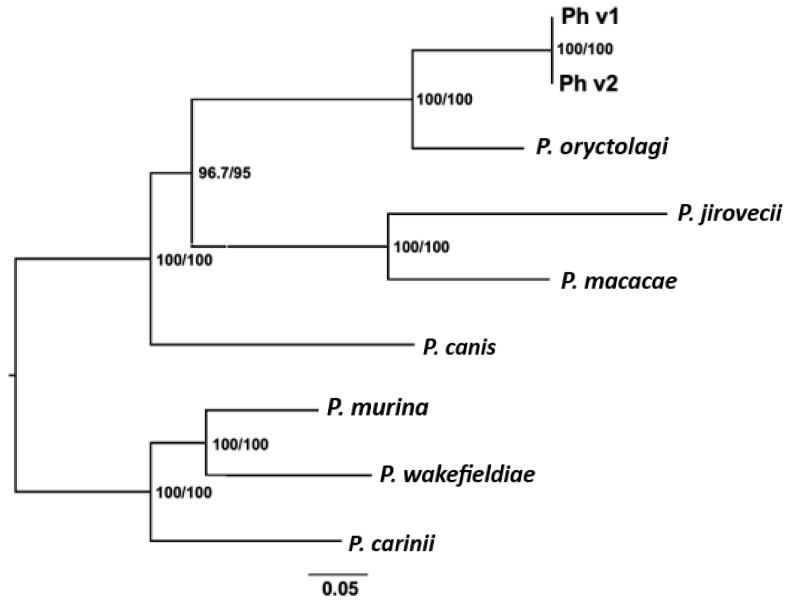
Maximum likelihood phylogenetic tree of *Pneumocystis* from hares and other mammals based on mitochondrial genome. The tree was constructed using concatenated nucleotide sequences of 15 protein-coding genes and 3 RNA genes from each mitochondrial genome (with GenBank accession numbers listed in Appendix A). Numbers at each node are SH-aLRT support (%)/ultrafast bootstrap support (%). The two *P.* sp. ‘*townsendii*’ variants identified in this study are represented by Ph v1 and Ph v2.

## Data Availability

All sequence data involved in this manuscript are available in NCBI’s Nucleotide Database, with accession numbers indicated in Tables and Figure legends, as well as Appendix A, particularly Appendix A.

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
