# Peer review of "Development of Highly Efficient Universal Pneumocystis Primers and Their Application in Investigating the Prevalence and Genetic Diversity of Pneumocystis in Wild Hares and Rabbits"

_jof, 2024, doi:10.3390/jof10050355_

Round 1
Reviewer 1 Report
1. An alignements of ITSs locus from different Pneumocystis species should be added
2. line 267-9: was ITSs and dhps PCRs also applied to the 14 Pneumocystis species ?
3. could a sudden shift of host species have occurred due to transient conditions to explain the proximity of Pneumocystis species present in lagomoprphs and primates ?
1. lines 46 to 52 : too long sentence
2. lines 54-55: this statement needs further explanations, as well as references
3. lines 58-59 : explain in what it differs : "ultrastrucutural morphology of P. oryctolagi differs from that of other Pneumocystis species"
4. line 130 : clarifiy how performance was evaluated
5. line 131 : primer Pu3Y.f misses onFIGs1, because it is amodification of Pu3for but this is mentioned only within discussion (line 460) so that it should be clarified in this figure
6. line 390 : the positions of the 4 codons should be given
7. line 408: "(Figure S5)" should be moved at the end of the previous sentence
Author Response
Reviewer 1:
Response to Comments for Authors
Major comments
- An alignements of ITSs locus from different Pneumocystis species should be
Response: As requested, the alignment has been added as Fig S3.
- line 267-9: was ITSs and dhps PCRs also applied to the 14 Pneumocystis species ?
Response: Only some, not all, of the 14 species were also tested by the ITS- and dhps-PCR. Since both the ITS and dhps loci are single-copy genes, they are expected to have a lower sensitivity than the multi-copy mtLSU and mtSSU genes as seen for the hare samples. Of the 91 and 6 hares positive for variants 1 and 2, respectively (by mtLSU/mtSSU-based PCR), only 33-55% were positive by ITS- or dhps-based PCR (lines 365-369). Due to their low sensitivity, we do not recommend using them for determing the prevalence of Pneumocystis, but recommend them for investigating genetic diversity and preliminary phylogeny analysis as noted in the manuscript (lines 493-495).
To further clarify this issue, we have revised the relevant information (lines 295-298).
- could a sudden shift of host species have occurred due to transient conditions to explain the proximity of Pneumocystis species present in lagomoprphs and primates ?
Response: Thank you for raising this valuable hypothesis. While we did not address it in this study due to unsuccessful whole nuclear genome sequencing of hare and rabbit samples as a result of their low Pneumocystis organism loads (unpublished observations), our previous phylogenomic analysis of P. oryctolagi and multiple other species has demonstrated that P. oryctolagi is more closely related to primate Pneumocystis than rodent Pneumocystis despite the general belief of the closer phylogenetic relationships of rabbits and rodents to each other than to primates. The observed phylogenetic discordance between P. oryctolagi and its host, the rabbit, suggests the occurrence of host switching events in their evolutionary past. Nonetheless, the evolutionary relationships among rodents, lagomorphs, and primates persist as a subject of controversy. There are multiple studies suggesting a closer relationship between lagomorphs and primates than with rodents as has been discussed briefly in the original manuscript. To further clarify, we have added the possibility of host shift to the revised manuscript (lines 566-567).
Detail comments
- lines 46 to 52 : too long sentence
Response: We have split it into shorter sentences (lines 46-52).
- lines 54-55: this statement needs further explanations, as well as references
Response: As requested, we have added more details along with references for Pneumocystis infection in rabbits (lines 56-62).
- lines 58-59 : explain in what it differs : "ultrastrucutural morphology of oryctolagi differs from that of other Pneumocystis species"
Response: As requested, we have added unique ultrastructural features of P. oryctolagi described in the literature (lines 63-67).
- line 130 : clarifiy how performance was
Response: Performance was estimated by visual inspection of the number, size and density of the DNA bands in the PCR products following agarose electrophoresis as shown in Figure 1. The best performance refers to the presence of a single, strong band with expected size. This information has been added to the Results section (lines 151-153).
- line 131 : primer f misses on FIG s1, because it is amodification of Pu3for but this is mentioned only within discussion (line 460) so that it should be clarified in this figure
Response: Primer Pu3Y.f is identical to Pu3for shown in Fig S1 except for the replacement of A at the 10th position (from the 5’ end) by a degenerate nucleotide R to compensate the variation between A and G. This information is indicated in Table S2. To further clarify, this information has also been added to the Fig S1 legend.
- line 390 : the positions of the 4 codons should be
Response: This information has been provided in Fig S5.
- line 408: "(Figure S5)" should be moved at the end of the previous
Response: Moved as requested.
Reviewer 2 Report
The manuscript by Ma et al. entitled “Development of Highly Efficient Universal Pneumocystis Primers and Their Application in Investigating the Prevalence and Genetic Diversity of Pneumocystis in Wild Hares and Rabbits” is very well written and precise.
Comprehensive research regarding Pneumocystis species is still sparse. Unravelling the distribution of Pneumocystis in the animal kingdom helps to understand the evolution of this fungus and how and why it causes such severe pneumonia in humans. It seems that the majority of the animal species examined thus far have been exclusively colonized by Pneumocystis, and this is observed in significant portions of their populations. Previous studies in humans have shown that Pneumocystis colonisation takes place at certain intervals in life but is not continuous. Nevertheless, most humans have Pneumocystis antibodies. However, information on Pneumocystis colonization and infection in animals are still lacking. A strict species-specificity is assumed or described in a few articles, but as there are also few data available on this, articles with studies such as the one presented here are even more important.
This article unravels another piece of this “Pneumocystis puzzle”. Studies on the prevalence in rabbits and hares are few and far between, and when carried out, only less sensitive methods such as stainings were used to analyze a small number of animals. The impressively high prevalence of Pneumocystis spp. in the animals examined and the presence of different genotypes were also shown in other animals, but not with detailed multigene analysis and not for hares and rabbits. The fact that these rabbit and hare Pneumocystis are more related to monkey and dog Pneumocystis than to other rodents is astonishing and should be investigated further.
The methods used are appropriate and comprehensive. They included multilocus analysis of mitochondrial and dihydropteroate synthase gene (DHPS) genes, as well as internal transcribed spacers (ITS).
In conclusion, this article makes an important contribution to the assessment of Pneumocystis prevalence in wildlife.
I only have few minor comments which should be best addressed with additional discussion points rather than additional work.
In Figure 3A, there should be differences in the DNA sequences of P.ory_v1 and P.ory_v2, but I cannot see any. They differ from the reference sequence, P.ory_ref, and all Hare_Ph sequences, of course, but differences among each other are not visible. Could it be that identical sequences have been inserted into the image by mistake? Please check and correct if necessary!
With conservative PCR analysis, as used in this study, a differentiation between Pneumocystis infection and Pneumocystis colonization is not possible, especially because the samples were also normalized prior the PCR application. I suggest including more quantitative methods (qPCR) in future projects. With qPCR you cannot differentiate between PCP and colonization directly, but you might get an idea of the Pneumocystis count per gram lung sample. It would also be great to include lung histology to identify lung infiltrates, infections or possibly a screening for other lung-related diseases in these animals. This may help to identify whether the animal suffered from a lung disease at time of the accident /dead / examination. Infections, as CMV infection, are known to be co-factors in developing a Pneumocystis pneumonia in humans. Therefore, the high prevalence of Pneumocystis spp. might also be an indicator for underlying diseases in these animals. Nevertheless, I completely understand that such analyses are not feasible in all cases.
It would be fair to mention that previous primer pairs were designed based on the earlier state of knowledge about Pneumocystis sequences. In the discussion, you criticise lightly the authors of earlier papers, but how could they have done better if information about Pneumocystis DNA sequences was lacking?
Pg 12, line 522: “similarly” instead of “simillarly”
Author Response
Reviewer 2
Major comments
The manuscript by Ma et al. entitled “Development of Highly Efficient Universal Pneumocystis Primers and Their Application in Investigating the Prevalence and Genetic Diversity of Pneumocystis in Wild Hares and Rabbits” is very well written and precise.
Comprehensive research regarding Pneumocystis species is still sparse. Unravelling the distribution of Pneumocystis in the animal kingdom helps to understand the evolution of this fungus and how and why it causes such severe pneumonia in humans. It seems that the majority of the animal species examined thus far have been exclusively colonized
by Pneumocystis, and this is observed in significant portions of their populations. Previous studies in humans have shown that Pneumocystis colonisation takes place at certain intervals in life but is not continuous. Nevertheless, most humans have Pneumocystis antibodies.
However, information on Pneumocystis colonization and infection in animals are still lacking. A strict species-specificity is assumed or described in a few articles, but as there are also few data available on this, articles with studies such as the one presented here are even more important.
This article unravels another piece of this “Pneumocystis puzzle”. Studies on the prevalence in rabbits and hares are few and far between, and when carried out, only less sensitive methods such as stainings were used to analyze a small number of animals. The impressively high prevalence of Pneumocystis spp. in the animals examined and the presence of different genotypes were also shown in other animals, but not with detailed multigene analysis and not for hares and rabbits. The fact that these rabbit and hare Pneumocystis are more related to monkey and dog Pneumocystis than to other rodents is astonishing and should be investigated further.
The methods used are appropriate and comprehensive. They included multilocus analysis of mitochondrial and dihydropteroate synthase gene (DHPS) genes, as well as internal transcribed spacers (ITS).
In conclusion, this article makes an important contribution to the assessment of Pneumocystis prevalence in wildlife.
I only have few minor comments which should be best addressed with additional discussion points rather than additional work.
Response: We deeply appreciate your expert acumen in the particular field pertinent to our study, and we are grateful for your insightful and encouraging remarks on our manuscript.
- In Figure 3A, there should be differences in the DNA sequences of ory_v1 and P.ory_v2, but I
cannot see any. They differ from the reference sequence, P.ory_ref, and all Hare_Ph sequences, of course, but differences among each other are not visible. Could it be that identical sequences have been inserted into the image by mistake? Please check and correct if necessary!
Response: Thsse two variants are identical in mtSSU shown in Fig 3A but different in mtLSU sequences (Fig S4) as noted in the end of the legend of Fig 3A. To further clarify, we have revised the legend by re-arranging the text and referring to Fig S4 for differences between these two variants.
- With conservative PCR analysis, as used in this study, a differentiation between Pneumocystis infection and Pneumocystis colonization is not possible, especially because the samples were also normalized prior the PCR application. I suggest including more quantitative methods (qPCR) in future projects. With qPCR you cannot differentiate between PCP and colonization directly, but you might get an idea of the Pneumocystis count per gram lung It would also be great to include lung histology to identify lung infiltrates, infections or possibly a screening for other lung-related diseases in these animals. This may help to identify whether the animal suffered from a lung disease at time of the accident /dead / examination. Infections, as CMV infection, are known to be co-factors in developing a Pneumocystis pneumonia in humans. Therefore, the high prevalence of Pneumocystis spp. might also be an indicator for underlying diseases in these animals.
Nevertheless, I completely understand that such analyses are not feasible in all cases.
Response: We appreciate your constructive and thoughtful comments. While we agree that quantitiation would be interesting, our primary goal in this study was to investigate the epidemiology of Pneumocystis infection rather than characterizing the intensity of infection and whether it represented Pneumocystis pneumonia or subclinical infection. Given the potential variation in the efficiency of amplification, we are also concerned that qPCR values may not be comparable among different species. We also agree that histopathology would definitely be of interest, and something we can consider in the future, but we are concerned that since many of the animals had died many hours or more prior to sampling of the lungs, autolysis and freezing/thawing would introduce artifacts and make interpretation unreliable as noted in the manuscript (lines 530-532). We will be describing lesions and diseases in this population in a separate manuscript that will highlight only severe and obvious lesions (manuscript in progress).To clarify these issues, we have revised the Conclusions accordingly (lines 585-589).
- It would be fair to mention that previous primer pairs were designed based on the earlier state of knowledge about Pneumocystis sequences. In the discussion, you criticise lightly the authors of earlier papers, but how could they have done better if information about Pneumocystis DNA sequences was lacking?
Response: We totally agree. It was never our intention to criticize or disparage the earlier studies and we have revised the relevant information accordingly (lines 476-478).
- Pg 12, line 522: “similarly” instead of “simillarly”
Response: Thanks for pointing the error. It has been corrected.